# New Genes in the *Drosophila* Y Chromosome: Lessons from *D. willistoni*

**DOI:** 10.3390/genes12111815

**Published:** 2021-11-18

**Authors:** João Ricchio, Fabiana Uno, A. Bernardo Carvalho

**Affiliations:** Departamento de Genética, Instituto de Biologia, Universidade Federal do Rio de Janeiro, Rio de Janeiro 21941-971, Brazil; rosaricchio@gmail.com (J.R.); fabi.uno@gmail.com (F.U.)

**Keywords:** *Drosophila willistoni*, Y chromosome, new genes, segmental duplication, gene duplication

## Abstract

Y chromosomes play important roles in sex determination and male fertility. In several groups (e.g., mammals) there is strong evidence that they evolved through gene loss from a common X-Y ancestor, but in *Drosophila* the acquisition of new genes plays a major role. This conclusion came mostly from studies in two species. Here we report the identification of the 22 Y-linked genes in *D. willistoni*. They all fit the previously observed pattern of autosomal or X-linked testis-specific genes that duplicated to the Y. The ratio of gene gains to gene losses is ~25 in *D. willistoni*, confirming the prominent role of gene gains in the evolution of *Drosophila* Y chromosomes. We also found four large segmental duplications (ranging from 62 kb to 303 kb) from autosomal regions to the Y, containing ~58 genes. All but four of these duplicated genes became pseudogenes in the Y or disappeared. In the *GK20609* gene the Y-linked copy remained functional, whereas its original autosomal copy degenerated, demonstrating how autosomal genes are transferred to the Y chromosome. Since the segmental duplication that carried *GK20609* contained six other testis-specific genes, it seems that chance plays a significant role in the acquisition of new genes by the *Drosophila* Y chromosome.

## 1. Introduction

Y chromosomes nearly always contain male-fertility genes, and in many species also carry the master sex-determining genes [1,2,3,4]. They usually contain a small number of genes but can be physically large due to the accumulation of repetitive DNA. For example, the *D. melanogaster* Y has ~40 Mbp and ~20 genes, whereas the X has ~42 Mbp and ~2300 genes [5]. The fact that X and Y pair and segregate during meiosis, as autosomal homologs do, suggests that the sex-chromosomes are also homologs and presumably derived from a regular pair of autosomes. The paucity of genes on the Y would be explained by massive gene losses (“degeneration”), as first suggested by Sturtevant and Muller (cited in [6]. Further theoretical and empirical studies (reviewed in [7,8]) supported these initial suggestions.

Briefly, the current view asserts that X and Y originated when one homolog from a regular autosomal pair acquires one or more sex-determining genes, becoming a proto Y, and its homolog become a proto X. A combination of evolutionary forces then favors the suppression of recombination between the proto-Y and the proto-X, allowing for further differentiation. The absence of recombination in the proto-Y reduces the efficacy of natural selection and ultimately leads to gene degeneration and loss, and to the accumulation of repetitive DNA [9], leading to the evolution of a mature Y chromosome. X chromosomes still recombine in females and suffer comparatively small changes in their gene content [10,11]. Mammals illustrate this canonical pathway well: their X and Y are derived from an autosomal pair, and the Y contains the master sex-determining gene *SRY* [12]. The human Y (the best-known case) encodes around 27 different proteins, 16 of which have close counterparts on the X, being relics of the ancient autosomal pair [3]. Hence, the hallmarks of the canonical model for the origins of Y chromosomes are ancestral X-Y homology and evolution by gene loss. Interestingly, three of the 27 genes of the human Y originated by duplication of autosomal or X-linked genes.

*Drosophila* Y chromosomes are also gene-poor but do not fit this pattern of relic X-Y homology and evolution by gene losses: all their known genes were acquired through gene duplications from the autosomes, and gene gains are ~10-fold more frequent than gene losses, at least in the period where data is available (in the last ~30–60 Mya [13,14]. These conclusions came mostly from the study of two species (*D. melanogaster* and *D. virilis*) in which Y-linked protein-coding genes were thoroughly identified [13,15,16,17,18]. The small number of well-studied Y chromosomes is not restricted to *Drosophila*: in general, Y chromosomes are poorly known because their characteristics thwart many genetic and genomic methods [19]. For example, Y chromosome sequences are usually fragmented in genome assemblies due to their richness of repetitive DNA and end up in a collection of unmapped fragments, which require specific methods for proper identification and assembly [13,20]. Exons of the same gene are frequently scattered in different scaffolds, making gene identification and annotation challenging. Hence it is not surprising that few Y chromosomes have been well studied.

It has been customary to refer to all gene movements to the *Drosophila* Y chromosome as “new genes” (e.g., [15,21]). In contrast, the field of new gene evolution adopts a more stringent definition that focuses on the creation of novelties (e.g., “we can define new genes as those that are present in all members of a monophyletic group but absent from all outgroup species” [22]). Simple gene movements have been called “positionally relocated genes (PRGs)” [23] and seem to be excluded from the above definition of “new genes”. Hence, it is relevant to ask if we should call the genes that move to the *Drosophila* Y “new genes”. We argue that the answer is yes because, for several reasons, these movements always entail significant biological novelty: (i) genes that move to the Y become completely invisible to selection in females; (ii) they lose recombination, because *Drosophila* males are achiasmatic and hence genes that move to the Y chromosome automatically fail to recombine; (iii) they move to a heterochromatic environment. These changes have important consequences: two distinctive features of *Drosophila* Y-linked genes—reduced codon bias and gigantic introns—most likely reflect the lack of recombination and the heterochromatic environment [24,25,26,27]. Furthermore, the study of *Drosophila* Y-linked genes and sensu stricto “new genes” share many features and concepts such as gene traffic and sex-antagonistic selection. Thus, it seems appropriate to call the genes that move to the *Drosophila* Y chromosome new genes.

The prominence of gene gains in *Drosophila* Y chromosomes makes them an attractive model to study the acquisition of new genes in Y chromosomes. We aim to address here two broad sets of questions: the “how” (i.e., the mechanics of acquisition of new genes by the *Drosophila* Y) and the “why” (e.g., the role of chance and natural selection). Regarding the first set of questions (“How”), acquisition of new genes in general occurs by de novo origin (e.g., from non-coding DNA) or by gene duplications. The available evidence strongly supports gene duplications as the primary mechanism in most species, and in *Drosophila* ~25% of them are mediated by an mRNA intermediate (“retro-transposition”), the remaining 75% being DNA-based duplications [28]. Anecdotal evidence [15,21,29] indicates that DNA-mediated duplication is also widespread among *Drosophila* Y-linked genes, but this issue has not been systematically studied until now. A related question is how exactly an ancestrally autosomal gene is “transferred” to the Y.

Much less is known about the second set of questions (the “Why”). Fisher [30] stated that “close linkage with sex may have enabled certain variants, beneficial in the male, to have established genetic stability, for, had they been autosomal, their deleterious effect in the female might have definitely outweighed their genetic advantages”. Putting this into the context of gene movements, one can say that Y chromosomes are expected to accumulate male-related genes because male–female antagonistic effects of genes may hamper their evolution unless they are located in a male-specific region of the genome. While this selective hypothesis could explain the strongly male-biased content of Y chromosomes, we should note that a neutral process can also do this, as follows (see also the SI in [14]). Gene duplications create genetic redundancy, which can be “resolved” by several mechanisms, including the degeneration of either the original or the new copy [31]. Suppose this happened with an autosomal or X-linked male-specific gene that duplicated to the Y: the result may be a gene transfer to the Y, even if there is no selection favoring Y-linkage. Now suppose that the gene that duplicated to the Y was a housekeeping or female-specific gene: selection will keep the original copy, since females need those genes, and the Y-linked copy will most likely be lost. Given that Y chromosomes have few genes and the autosomes and X have many, the expected outcome of the above mechanisms over evolutionary time is a “gene traffic” of male-specific genes from the other chromosomes to the Y, even if there is no immediate fitness advantage for these male-specific genes being Y-linked. Which of these two hypotheses for the accumulation of male genes in the Y chromosomes (natural selection or chance) is correct, or more prevalent, is a hard question. Our aim here is necessarily humbler: to call attention to this question and provide some elements of the answer (namely, the role of chance events).

Given the small number of well-studied *Drosophila* Y chromosomes, adding new species is a worthwhile effort. In this paper we describe the identification of the Y-linked protein-coding genes of *D. willistoni*. As *D. melanogaster*, *D. willistoni* belongs to the *Sophophora* subgenus; they diverged shortly after the split between the *Sophophora* and *Drosophila* subgenera (30–60 Mya; [32,33]). Hence their divergence time is quite large, similar to their divergence from *D. virilis*, which belongs to the *Drosophila* subgenus.

## 2. Materials and Methods

### 2.1. Fly Strains and Genome Assemblies

Unless otherwise noted, we used the reference *D. willistoni* strain 14030-0811.24, which was sequenced as part of the 12 *Drosophila* Genomes Project [34]. This strain was originally collected in the Guadeloupe Island (Caribbean) and belongs to the *D. willistoni willistoni* subspecies [35]. We used the reference assembly (GCA_000005925.1). This Sanger assembly came from DNA collected from unsexed embryos [34] and hence contains the sequences from the Y chromosome. We also used two recent long read assemblies. Assembly GCA_018902025.1 employed PacBio technology; the current NCBI info states that it came from females, but the long reads actually came from males (José Ranz, personal communication), and indeed we found all Y-linked genes there. Assembly GCA_018903445.1 is based on Nanopore technology [36] and used male DNA from strain 14030-0811.17, which was collected in Uruguay, and hence belongs to the subspecies *D. willistoni winge* [35]. The genomes of *D. paulistorum* (two different strains), *D. equinoxialis* and *D. tropicalis* came from [36]. These species are closely related to *D. willistoni*, and we used their genomes as outgroups in several analyses.

### 2.2. Illumina Reads and Testis Transcriptome Assembly

As detailed below, Y-linked scaffolds can be identified by comparing an assembled genome with Illumina reads from genomic DNA of males and females. We used the accessions SRR9426110 and SRR9426117 for this purpose. To estimate gene expression we used RNA-seq reads from different organs and body parts (accessions listed in Appendix A). A de novo testis transcriptome is very helpful in annotating Y-linked genes; we used the assembly GJOF01000000. The testis transcriptome and all Illumina reads came from the reference strain 14030-0811.24.

### 2.3. Identification of Y-Linked Scaffolds and Annotation of Y-Linked Genes

We used the YGS method to computationally identify the Y-linked scaffolds by comparing the assembled genome with male and female Illumina reads [13]. This procedure identifies Y-linked sequences but not the genes they contain. For example, in *D. virilis,* most Y-linked scaffolds contain only repetitive DNA and other non-coding sequences [13]. In order to identify and annotate the protein-encoding genes we did blast searches in the Y-linked sequences identified by the YGS method against databases of Dipteran proteins, bacterial and yeast proteins (to identify contaminant scaffolds), and transposable elements. These standard procedures were applied considering that, in many Y-linked genes, exons of the same gene are scattered in different scaffolds due to assembly failure in long, repeat-rich introns [29]. Similar challenges are met during the annotation of heterochromatic genes in general [37,38]. Furthermore, Y-linked genes frequently have gaps due to low coverage in Sanger reads: the reference genome of *D. willistoni* was sequenced at 8.4× depth from unsexed embryos, so the expected coverage of the Y chromosome is 2.1×. We fixed the fragmentation and gaps in the candidate genes by comparing the assembled genes with the testis transcriptome assembly. We detected both functional genes and pseudogenes during annotation. We adopted two criteria to classify them: functional genes have intact CDS (without premature stop codons and frame-shift indels) and significant expression at the RNA level (defined here as contributing more than 10% of the functional product). Y-linkage of all candidate genes was confirmed by PCR using male and female DNA.

### 2.4. Timeline of Gene Acquisitions by the Y Chromosome

The Y-linked genes found in *D. willistoni* might have been acquired at any point in the phylogeny. We identified this point by examining species progressively more distant from *D. willistoni* (within the *willistoni* group) and going up to *D. saltans* and *D. melanogaster.* For the genes transferred to the Y chromosome (i.e., no autosomal copy present), we used standard PCR in the other species with primers designed with the *D. willistoni* genes. In cases of amplification failure, we repeated the PCR using degenerate primers ([39]; Appendix A). Recent duplications to the Y which left an intact autosomal copy cannot be reliably mapped outside *D. willistoni* with PCR because the few mismatches we used to design Y-specific primers are not guaranteed to have persisted across species. In these cases we used the following preliminary procedure. We performed blastN searches using as queries the *D. willistoni* Y-linked genes and as databases the recently sequenced male genomes of *D. paulistorum*, *D. equinoxialis*, *D. tropicalis*, *D. nebulosa*, and *D. saltans* [36]. If we find only one hit in a large scaffold (>~5 Mbp) which contains many other genes (and hence is euchromatic), we safely conclude that the gene was not duplicated in the target species. In contrast, two hits (one in a large scaffold, the second in a smaller one) suggest that duplication to the Y had already occurred in the target species.

### 2.5. Measurement of Gene Expression

We measured gene expression of the Y-linked and some other genes using the RSEM program [40] and RNA-seq databases from different body parts and organs of *D. willistoni* (Appendix A). Y-linked genes are frequently misannotated due to the assembly problems mentioned above, so we needed to build a custom reference transcriptome. We started with the FlyBase annotation release 1.3 (downloaded in 12 October 2011from http://ftp.flybase.net/genomes/dwil/dwil_r1.3_FB2010_02/fasta/dwil-all-CDS-r1.3.fasta.gz), and removed from it all sequences matching the correctly annotated 22 *D. willistoni* Y-linked genes (the eight previously known, plus those identified in the present paper), using the *cd-hit-est-2d* program [41]. We then added back the correctly annotated 22 Y-linked genes and ran the RSEM program.

### 2.6. Detection of Segmental Duplications in the Y Chromosome

We separately aligned male and female Illumina genomic reads (accessions SRR9426110 and SRR9426117) against the reference Sanger assembly using the *bwa* software [42], and obtained coverage estimates using the *bedtools* package [43]. We then inspected coverage in suspicious scaffolds (which have many pseudogenes), looking for increases in coverage. We obtained the exact coordinates of the duplications by examining the confirmed duplications with the IGV browser [44], looking for increases in coverage and presence of male-specific SNPs.

## 3. Results

### 3.1. Identification of Y-Linked Sequences of D. willistoni

Figure 1 shows the results of the YGS method as applied to the reference Sanger assembly (top row) and a PacBio assembly (bottom row) of the same strain of *D. willistoni*. The bimodal distribution separates the Y-linked sequences (peak on the right-hand, with a large amount of sequence unmatched by female reads) from the X and autosomes (left-hand peak). The red dots mark the scaffolds matching the eight previously known Y-linked genes of *D. willistoni* [13,14]. All these sequences are located in the right-hand peak (average for Sanger assembly: 94%; range: 87–100%); they served as a positive control and show that the YGS method reliably identifies Y-linked sequences in *D. willistoni*. In the intermediate region of the graph, between the two peaks, there are numerous small sequences in the Sanger assembly; they correspond to misassembled repetitive regions [13]. YGS results are usually presented as in the first two columns of Figure 1, which may mislead the reader to conclude that Y-linked sequences represent a large proportion of the genome. The last column, which shows the amount of sequence in the scaffolds, clarifies that the Y chromosome is relatively small; because it is a repeat-rich region, Y-linked sequences are prone to assembly fragmentation and tend to get scattered in numerous small scaffolds. If we adopt the cut-off of 60% unmatched sequences, we find that the Y-chromosome in the Sanger assembly contains 28.5 Mbp (in 9031 scaffolds), whereas the X and autosomes amount to 189.8 Mbp (in 2107 scaffolds); the corresponding values for the PacBio assembly are 49.4 Mbp (in 605 scaffolds) and 197.6 Mbp (in 139 scaffolds). As suggested by the sharp bimodality shown in Figure 1 (panels **c** and **f**), the above estimates are quite robust in relation to the 60% cutoff; adopting an 80% cut-off would reduce the estimated size of the Y chromosome to 26.7 Mbp (Sanger assembly) and 48.6 Mbp (PacBio assembly). PacBio assemblies tend to be larger because they have a better representation of repetitive sequences; note that assembled size of the Y chromosome (being repeat-rich) increased much more than that of the X and autosomes. Another advantage of PacBio and Nanopore assemblies compared to Sanger or Illumina is that they are much less fragmented: the Sanger assembly contains 14,838 scaffolds, the PacBio contains 748, and the Nanopore contains 336. The smaller fragmentation helps the identification of Y-linked scaffolds: in the Sanger assembly, 3700 scaffolds, amounting to 5.3 Mbp, could not be classified by the YGS method because they are not informative (contain less than 20 valid single-copy k-mers; [13]); the corresponding figures for the PacBio assembly are four scaffolds and 71 kb. Despite these advantages, we will use the Sanger assembly in most analyses because it is the reference assembly.

### 3.2. Identification of Y-Linked Protein Coding Genes

We found all eight previously known Y-linked genes of *D. willistoni* while annotating the 28.5 Mbp (9031 scaffolds) predicted to be Y-linked (their orthologs in *D. melanogaster* are: *kl-2*, *kl-3*, *PRY*, *PPrY*, *ORY*, *CCY*, *ARY*, and *JYalpha*; [13,14]. Besides these, we found 14 protein-coding genes, reported here as Y-linked genes for the first time (Table 1).

We confirmed Y-linkage of these genes with PCR in male and female DNA (e.g., [39]); no false-positives were found. Eleven out of these 14 genes have been automatically annotated during the sequencing of the *D. willistoni* genome [34], and the remaining three by Yang et al. [45]. None of these have been manually curated before (Appendix A). We adopted a conservative nomenclature by keeping their original names, even if only a small portion of their coding sequence had been identified before. Due to the gaps and fragmentation in the genome assembly in most cases the original annotation was incomplete. In the *Piezo-like* ortholog parts of the gene were annotated as three different genes (*GK27406*, *GK18657*, and *GK27211*; we kept the name *GK27406*, which is the largest gene). A similar situation happened with *YOgnWI000172*. The main problems we found during gene annotation were the fragmentation and the gaps, but the genomic sequence of the multicopy genes *GK20618*/*GK20619* and *GK18510* may have mis-assemblies. Due to these uncertainties the classification of *GK18510* as a functional gene should be seen as a preliminary result (Appendix A).

All 14 genes were acquired by gene duplications from ancestrally autosomal or X- linked genes which already had testis-specific expression (Table 1 and Appendix A). The evidence of gene duplication is unequivocal for recent movements because, in these cases, we still found the original autosomal copy, either as a functional gene (as occurred with *GK18510*, *GK20591*, *YOgnWI018045*, and *GK20619*) or as a pseudogene (as occurred with *GK20609* and *GK21220*; see Discussion). For older acquisitions, the evidence is based on parsimony. For example, as detailed in Section 3.5, the Y-linked gene *GK27406* has autosomal orthologs in five progressively more distant outgroups (*D. insularis*, *D. nebulosa*, *D. sucinea*, *D. saltans,* and *D. melanogaster*), so its ancestral location must be autosomal. Its Y-linkage in *D. willistoni* must be the result of one gene duplication to the Y, followed by the loss of the original autosomal copy. The alternative hypothesis (ancestral Y-linkage) would imply five duplications, coupled with the loss of the original Y-linked copy in each case (see ref. [14] for details).

The pattern of gene duplications from autosomal (or X-linked) testis-specific genes we observed in *D. willistoni* fits what is found in other *Drosophila* species, except that *D. willistoni* is the first species in which X-linked genes were found as the source of duplications (we confirmed the X-linkage by looking at the *D. paulistorum* genome; see below). Ancestrally X-linked genes that move to the Y have potential implications for hybrid sterility, on which we will comment further in the Discussion. The 14 genes span a range of functions, such as enzymes and structural proteins of the spermatozoon. The first 10 genes showed in Table 1 were fully transferred to the Y, in the sense that the original autosomal copy was lost or reduced to a pseudogene. Therefore, they will most likely stay in the Y chromosome. The fate of the last four genes (*GK20591*, *YOgnWI018045*, and the multicopy *GK20618*/*GK20619* and *GK18510*) is less certain because a functional autosomal copy remained at the original location (the genes *GK19651*, *GK14595*, *YOgnWI012342*, and *GK18077*, respectively), and both copies have a similar level of expression (Appendix A; Appendix A). The high similarity between their Y and autosomal copies (equal or greater than 96% identity at the nucleotide level) implies a recent origin (see section “When did the *D. willistoni* Y chromosome acquire its genes?”). It seems that, for these four genes, all outcomes are possible: loss of Y-copy, loss of autosomal copy, or retention of both, perhaps via subfunctionalization [31]. Some of them, in particular the multicopy with pseudogenes *GK18510* and *GK20618*/*GK20619,* may be examples of what Betrán [46] called ‘‘live fast, die young’’: genes that participate in arms races such as male–male competition or male–female antagonism, and that do not last long in the genome.

Besides functional genes, *Drosophila* Y chromosomes contain a number of pseudogenes, as happens in other heterochromatic regions [13,15,37,47,48]. As detailed in the Material and Methods, we used the intactness of the coding regions and the expression at the RNA level as criteria to sort pseudogenes from functional genes. Such classifications may be too arbitrary, particularly if there is a large gray zone. This was not the case in *D. willistoni*. Besides the *GK18510* gene mentioned above (and detailed in Appendix B), the two most doubtful cases we found are: (i) the annotated “gene” *GK22179* (which has an expression level of ~3% of its autosomal counterpart, *GK11098*), and (ii) an unannotated sequence in scaffold CH961348 (similar to the autosomal gene *GK11572*), which most likely was an artifact in the Sanger assembly: PCR repeatedly fail to amplify it in males and females, and it is absent from the PacBio assembly. We found 45 pseudogenes (Appendix A), by far the largest number ever found in a *Drosophila* Y chromosome [13,15,37,47,48]. This finding has an interesting explanation: the *D. willistoni* Y chromosome harbors four large segmental duplications from autosomes, which originally contained ~58 protein-coding genes. All but four became Y-linked pseudogenes or disappeared.

### 3.3. Y-Linked Segmental Duplications from Autosomes

During the annotation of Y-linked scaffolds it came to our attention that most pseudogenes have ancestral locations close to each other (Appendix A), suggesting that they originated from segmental duplications of autosomal or X-linked chromosomal regions. We confirmed this suspicion by looking at the coverage across these regions (Figure 2; [49]). We identified four large segmental duplications (Table 2), and there are probably additional smaller ones.

The very high coverage in these regions shows that they have multiple copies inside the Y chromosome (in some cases ~10 copies), and the unevenness of the coverage implies that they suffered partial deletions and duplications. Given the high copy number, it is not surprising that this region is heavily fragmented in the Sanger assembly (and to a lesser extent in PacBio). Indeed, among the 9031 Y-linked scaffolds that we found in the Sanger assembly, 3033 match these four regions (the matching regions span ~1.7 Mbp). This duplication to the Y, followed by additional duplication and deletions, is reminiscent of the *FDY* region found before in the *D. melanogaster* Y [15], except that its 11 kb are dwarfed by the size of the *D. willistoni* segmental duplications.

These segmental duplications created four functional genes (*GK20609*, *YOgnWI018045*, and the multicopy genes *GK20618*/*GK20619* and *GK18510*), while ~54 became pseudogenes or disappeared. This finding raises the question of what factor(s) influenced the fate of these duplicated genes. The autosomal or X-linked ancestors of all four surviving genes are testis-specific genes, as judged by their orthologs in *D. melanogaster* (Table 1) and their gene expression in *D. willistoni* (Appendix A). The same pattern has been previously found in the well-studied *D. melanogaster* and *D. virilis* [13,16]. Hence, testis-specific expression is necessary for the retention of a gene in the *Drosophila* Y chromosome. However, it is not sufficient: among the 54 lost genes, 10 have testis-specific expression in *D. willistoni*, and in most cases the same is true for their *D. melanogaster* orthologs. We will return to this point in the Discussion

### 3.4. Mechanism of Duplications to the Y: DNA or RNA-Based?

In order to identify the duplication mechanism, we used two criteria. First, genes that are part of a segmental duplication certainly originated through a DNA duplication. Second, we looked for conserved intron positions between the Y-linked gene, and its autosomal (or X-linked) ortholog in another species, since RNA-mediated duplications lead to intron loss. We initially used only *D. melanogaster* as the outgroup. However, we found one case (the *GK20609* gene) suggesting complete intron loss in a gene duplicated as part of a segmental duplication, which does not make sense. It is known that the *D. willistoni* lineage underwent many intron losses [50], which are unrelated to the movement of genes. We circumvented this problem by looking at the intron–exon structure in the species of the willistoni group that have been recently sequenced by Kim et al. [36]. In these species, the *GK20609* ortholog is autosomal and already intron-less. We cannot ascertain the duplication mechanism for seven out of the 14 genes (they are intron-less in the outgroup species and are not part of a segmental duplication). The remaining seven genes were DNA-based duplications: three have conserved intron positions (*GK21041*, *YOgnWI000172*, and *GK27406*), and four are part of segmental duplications (*GK20609*, *GK18510*, *YOgnWI018045*, and *GK20618*/*GK20619*).

### 3.5. When Did the D. willistoni Y Chromosome Acquire Its Genes?

Previous investigations found eight genes in the *D. willistoni* Y by searching for genes found to be Y-linked in *D. melanogaster* and *D. virilis* [13,14]. Thus, almost by definition, these genes were acquired before the split between *D. willistoni* and these two species. In this paper, we used a de novo identification approach (YGS), so the Y chromosome might have acquired its genes at any point in the phylogeny. We investigated this issue as detailed in ref. [39] and in the Material and Methods. Similar to what has been seen in previous studies [13,14], we found that the *D. willistoni* Y-linked genes were acquired throughout the phylogeny (Figure 3; Appendix A).

We also extended this investigation to the subspecies level. *D. willistoni* contains three subspecies with partial reproductive isolation [35,52]. Two of them have been sequenced in males: *D. w. willistoni* (to which the reference strain belongs) and *D. w. winge* [36]. We found that all 22 Y-linked genes (the eight previously known plus the 14 identified in the present work) are Y-linked in the *D. w. winge* strain (i.e., they are located in the right-hand peak of the YGS graph). Hence, they were acquired by the Y chromosome before the split of these two subspecies.

## 4. Discussion

Previous work searched in *D. willistoni* for protein-encoding Y-linked genes in two other *Drosophila* species with well-known Y chromosomes, *D. melanogaster* and *D. virilis*. These efforts identified eight genes [13,14]. Here, we used a de novo approach and found 14 additional genes, all acquired after the split between *D. willistoni* and these two species. Below we discuss the main implications of these findings.

### 4.1. D. willistoni Y-Linked Genes: Comparison with Other Drosophila Species

The Y-linked gene content of *D. willistoni* shares several important features with the two other species with well-known Y chromosomes, *D. melanogaster* and *D. virilis*. First, in the three species nearly all genes originated by duplications from genes already strongly or exclusively expressed in testis. The only known exception is *FDY*, a recent Y-linked gene from *D. melanogaster* whose ancestral gene (*vig2*) is expressed in many tissues and organs (including testis) and is strongly expressed in ovaries [53]. Second, the gene duplications to the *D. willistoni* Y were mediated by a DNA mechanism in all seven cases that we could ascertain. This probably holds true for other *Drosophila* species, although they have not yet been systematically examined in this respect. If confirmed, this would be an interesting difference between genes acquired by the Y and the other chromosomes, which involve an RNA intermediate in 25% of the cases [28]. A possible explanation for the absence of RNA-based duplications to the *Drosophila* Y is that this chromosome (as other heterochromatic regions) is a harsh environment for arriving genes: it has been long known that euchromatic genes that move to heterochromatic regions are silenced [54], and it is likely that a gene duplication carrying flanking euchromatic sequence has a higher survival chance than a naked, promoter-less retrocopy. The third commonality among the three species is the preponderance of gene gains over gene losses, as suggested by the inspection of Figure 3 and confirmed by the statistical analysis. As detailed in the Appendix A, using the data of the three species we found a gain–loss ratio of 25 (*p* = 0.002; 95% confidence interval: 3.4–184.5), and the same qualitative result is obtained when removing the four genes whose original autosomal copies are functional (Appendix A and Appendix A). These four genes—*GK20591*, *YOgnWI018045*, and the multicopy *GK18510* and *GK20618*/*GK20619*—arguably can become pseudogenes in the Y; if we remove them, we obtain a gain–loss ratio of 21; *p* = 0.003. It will be interesting to look at other species, particularly outgroups, to better understand what is happening. However, it is already clear that *Drosophila* Y chromosomes are not evolving according to the canonical theory of Y chromosome evolution (see Appendix B for an alternative view).

On the other hand, the *D. willistoni* Y chromosome has some features not seen in other *Drosophila* species. First, it seems that a recent burst in gene acquisitions happened after the split between *D. willistoni* and its cryptic species *D. paulistorum*/*D. equinoxialis*, which created a set of ~10 private Y-linked genes in *D. willistoni*. This gene gain burst is even more remarkable given the rather short time interval (4.8 Mya; Ref. [55]) and the lack of recent bursts in the well-studied *D. melanogaster* and *D. virilis* (the former has one or two private Y-linked genes [15,56]; *D. virilis* has none). Second, it seems that the gene gain rate is higher in *D. willistoni*. The previous estimates of the gain-loss ratio were 10.7 and 4.9 (using *D. melanogaster* and *D. virilis* as the focal species, respectively [13,14], and when we included *D. willistoni* we got 25. Furthermore, the heterogeneity in the gain–loss ratio among branches becomes statistically significant, suggesting that *D. willistoni* is an outlier. Third, we found four large segmental duplications that copied ~700 kb of autosomal sequence in the *D. willistoni* Y chromosome (Table 2); the previously known cases have a few kb [15,56]. The first and second peculiarities of the *D. willistoni* Y may be a consequence of the third: a higher rate of segmental duplications to the Y is expected to increase the gene gain rate and, if recent, may generate a fairly large amount of recent gene acquisitions by the Y. We got mixed results while trying to find evidence for this hypothesis in the literature. Suppose this hypothesis is correct and that segmental duplications occur in other chromosomes as well. In that case, one might expect to find increased gene movements in general, but *D. willistoni* does not seem to be an outlier in this respect [23]. On the other hand, Vibranovski et al. [28] analyzed the same dataset of [23] by partitioning it in A→A, A→X, and X→A movements and noted that *D. willistoni* is an outlier to the general pattern of excess of gene movements out of the X: it has many A→A and A→X movements, which weakly supports the idea that gene gains are more frequent in the *D. willistoni* lineage. Perhaps at this point, the most solid conclusion we can obtain was already outlined in the 12 *Drosophila* Genome Project paper: “*D. willistoni* is an exceptional outlier by several criteria, including its unusually skewed codon usage, increased transposable element content and potential lack of seleno-proteins” [34,57]. The list goes on: “Some clades, like the *willistoni* group, seem to undergo many more [intron] losses per million years than others” [50]. It will be interesting to directly investigate the occurrence of segmental duplications in the X and autosomes to verify if it is indeed increased in the *D. willistoni* lineage and if these duplications created new genes. In particular, one can look at the unexpected gene movements reported by Vibranovski et al. [28], and check if several genes came from adjacent chromosomal regions, which is a telltale sign of a segmental duplication. Finally, careful studies of more *Drosophila* Y chromosomes might help us better understand the relationship between segmental duplications and gene traffic to the Y chromosome.

### 4.2. Gene Movements to the Y Chromosome and Reproductive Isolation

All previously known *Drosophila* Y-linked genes originated from duplications of autosomal genes, whereas two *D. willistoni* Y-linked genes were formerly X-linked (Table 1). In itself, this may be just a coincidence: the sample size is small (there were seven gene acquisitions in the *D. melanogaster* lineage whose ancestral location is known, and four in *D. virilis* [13,14]), and male genes (which usually originate Y-linked genes) are underrepresented in the X [10,11]. Nevertheless, X-Y gene movements potentially have major biological consequences on speciation. Namely, if a gene essential for male fertility moves from the X to the Y in one species (or population), it instantly creates unidirectional hybrid male sterility (i.e., one direction of the interspecific cross will produce sterile F1 males, and the other will produce fertile F1 males). Unidirectional hybrid male sterility is indeed quite common in *Drosophila* [58]. The most relevant case here is the recent Y-linked gene *GK28041* whose orthologs are X-linked in *D. melanogaster* (ortholog: *CG32650*) and *D. paulistorum* (it is flanked by Muller A genes in the *D. paulistorum* assemblies reported in [36]; the gene order is: *CG4542-CG32649-CG32650-Pde9-CG3775*). Crosses between *D. willistoni* and *D. paulistorum* yield unidirectional hybrid male sterility, but (disappointingly) in the “wrong” direction: *D. willistoni* male × *D. paulistorum* female produce sterile males, and the reciprocal cross produces fertile males [59]. So *GK28041* is not essential for male fertility, and other sterility factors play the major role in those hybrids. Another interesting case is provided by three subspecies of *D. willistoni*, in which hybrids also display unidirectional hybrid male sterility [35,52]. As commented before, all 22 genes (the eight previously known plus the 14 identified in the present work, including *GK28041*) are Y-linked in the two sequenced subspecies. Hence, X–Y movements of the known Y-linked *D. willistoni* genes are not the cause of hybrid sterility in crosses between these two subspecies. While both cases we investigated here yield negative results, gene movements can cause hybrid sterility, as shown by Masly et al. [60], and are a particularly powerful mechanism when they occur between the X and Y.

### 4.3. How Do Male Genes Move to the Y Chromosome?

In most *Drosophila* Y-linked genes, the only sign of their autosomal (or X) origin is their location in other species. The recent acquisition of the *FDY* (flagrante delicto Y) by the *D. melanogaster* Y chromosome left more clear signs of this process: an 11 kb autosomal sequence containing five genes got duplicated to the Y, and all genes except *FDY* (which is a copy of *vig2*) became pseudogenes, their sequences becoming scrambled by deletions, duplications and point mutations [15]. However, in *FDY*, the autosomal copy is active, so this gene does not represent the most common case of a gene transfer to the Y chromosome. Two *D. willistoni* genes provide snapshots of this transfer process.

The segmental duplication involving scaffold CH964272 (Table 2) duplicated a 265 kb long autosomal sequence, which originally contained 25 genes, to the Y chromosome. Remnants of 12 genes (Appendix A), and one functional gene, *GK20609* (Table 1) are found in the Y chromosome. Interestingly we found a mirrored situation when we looked at the autosomal sequence: all genes are functional, except for a pseudogenized copy of *GK20609*, that contains many deletions and substitutions (Figure 4). This provides direct evidence of how male genes move to the Y. First, an autosomal or X-linked gene duplicates to the Y (possibly as part of a segmental duplication). Then, in some cases, the genetic redundancy is “resolved” by the degeneration of the original copy; over time, the pseudogene in the autosome (or X), and its flanking sequences in the Y chromosome tend to disappear. Eventually, a functional Y-linked gene whose ortholog is autosomal or X-linked in other *Drosophila* species is the only remaining sign of the original gene duplication.

The second example we found caught a later stage of the above process. Blast searches with the Y-linked gene *GK21220* found the Y-linked scaffolds (as expected) and a very similar 118 bp match in the autosomal scaffold CH963850 (coordinates 5158275–5158392). This small matching region is also present in the PacBio and Nanopore assemblies, ruling out assembly error. When we inspected the corresponding region in the *D. paulistorum* assembly [36], we found that it is autosomal (as expected) and contains the functional ortholog of the *GK21220* gene (Appendix A). This shows that the 118 bp region of scaffold CH963850 in the *D. willistoni* genome is a remnant of the original *GK21220* gene, which degenerated after its duplication to the Y. We could not find any sign of the flanking genes in the *D. willistoni* Y, either because they were not copied to the Y, or had degenerated beyond recognition.

### 4.4. Why Do Male Genes Move to the Y Chromosome?

The traffic of male genes to the Y chromosome (e.g., Figure 3) intuitively suggests that natural selection favors the Y-linkage of these genes. However, as argued in the Introduction, this is not necessarily correct. The Y-linked segmental duplications of *D. willistoni* are very useful in this respect because they provide replicas: many genes were transferred at the same time to the same location in the Y chromosome; a few of them became functional Y-linked genes, and most were lost. Testis-expression seems to be a necessary condition for the establishment of a functional gene in the *Drosophila* Y chromosome. Still, as shown by the *D. willistoni* data, it is not sufficient: among the 25 genes duplicated to the Y by the scaffold CH964272 segmental duplication, seven were testis-specific (*GK13044*, *GK13038*, *GK14208*, *GK13026*, *GK26902*, *GK28338*, and *GK20609*). Only *GK20609* survived as a functional gene; the remaining six became Y-linked pseudogenes or disappeared. Why?

A possible selective explanation would be gene amplification: Y chromosomes (as other heterochromatic regions) are prone to accumulate duplications [3,15,56], and if natural selection favors multiple copies for some male genes, there may be an advantage in transferring to the Y chromosome, due to several mechanisms. For example, the gene may be more easily duplicated, or the duplications inside the Y may be more easily tolerated because there is no recombination (and hence ectopic recombination is greatly reduced), or because the Y has very low gene density (and thus there is a smaller chance of interfering with other genes). While there are examples of gene amplifications in *Drosophila* and human Y chromosomes Y [3,61], and two recent Y-linked genes of *D. willistoni* are multicopy (*GK18510* and *GK20618*/*GK20619*), we can rule this out as a general explanation for *Drosophila* because nearly all known *Drosophila* Y-linked protein-encoding genes (including *GK20609*) are single copy [13,16,17,18]. Another popular selective explanation is that male genes may evolve more easily on the Y chromosome due to antagonistic selection in females [30]. While we cannot reject this explanation, we must note that it is not compelling either: *Drosophila* Y-linked genes have a myriad of functions (motor proteins, structural proteins of the sperm, enzymes) whose only common feature seems to be testis expression. If antagonistic selection in females plays a significant role in this process, we must conclude that it was present in the diverse set of genes that successfully moved to the Y (e.g., *GK20609*) and absent in those that failed (e.g., *GK13044*, *GK13038*, *GK14208*, *GK13026*, *GK26902*, *GK28338*; Figure 4).

Alternatively, chance might play a significant role in the “choice” of which gene copy (the original or the new Y-linked one) stays functional and which one becomes a pseudogene. Of course, chance does not imply 50% probability, and indeed in the CH964272 segmental duplication, only one out seven testis genes that duplicated to the Y eventually became established there. Given that Y chromosomes have few genes and the autosomes and X have many, this mechanism by itself would create gene traffic of male genes from the X and autosomes to the Y chromosome. While we cannot at this moment prove that this hypothesis is correct, it seems advisable to consider it, along with natural selection-based models, as possible explanations for the male gene traffic to the Y chromosome. Finally, it seems to us that the above discussion does not apply to the X-autosome gene traffic problem: unlike the Y chromosome, the X is neither restricted to one sex nor extremely gene-poor.

### 4.5. Concluding Remarks and Future Directions

There is much we do not understand about *Drosophila* Y chromosomes, and few of these chromosomes have been studied in detail. As a consequence, each newly studied species brings new aspects to light. The case of *D. willistoni* highlighted the role of segmental duplications in the origin of new genes in the Y chromosome and possibly also in other chromosomes. Besides this, the segmental duplications create replicas to study the evolutionary fate of genes that are duplicated to the Y, which opens a window to investigate the roles of natural selection and chance in the gene traffic to the Y. Further studies are likely to bring additional surprises, particularly in areas that have been barely touched, such as *Drosophila* outgroups [17] (e.g., how far in the phylogeny the pattern of “more gains than losses” goes) and non-coding RNA [62] (which may help explain the manifold effects of *Drosophila* Y chromosomes in phenotypes such as longevity [63], gene expression [64], and hybrid sterility [65]).

## Figures and Tables

**Figure 1 genes-12-01815-f001:**
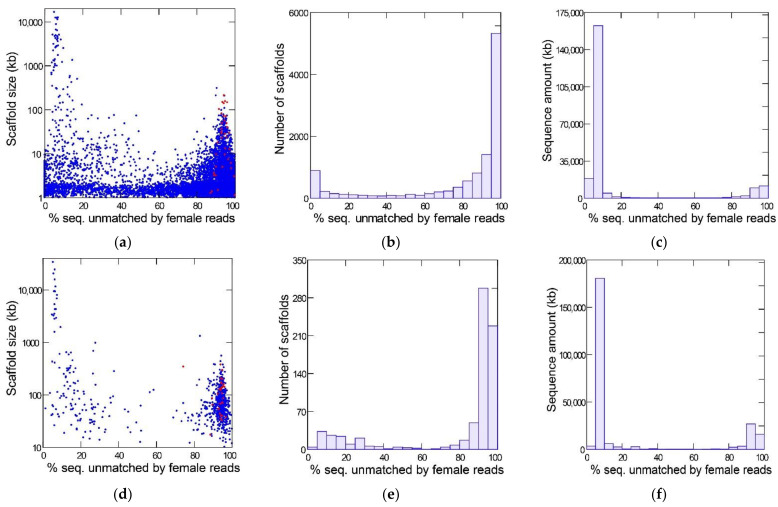
Detection of Y-linked sequences in the reference *D. willistoni* genome with the YGS method. (**a**–**c**): reference Sanger assembly; (**d**–**f**), PacBio assembly. The *X*-axis in all graphs is the evidence of Y-linkage; scaffolds with a high proportion of sequence unmatched by female reads are Y-linked. In the left-hand panels (**a**,**d**), each dot represents a scaffold, and the red dots mark the previously known Y-linked sequences. The center panels (**b**,**e**) show the scaffold distribution as the number of scaffolds, and the right-hand panels (**c**,**f**) show the scaffold distribution as the sequence amount. Note that the bulk of the sequence can be reliably classified as Y-linked (peak on the right-hand) or X/autosomal (left-hand peak). Non-informative scaffolds (less than 20 single-copy *k*-mers) were excluded from the analysis (see text). The raw YGS results are available in the Appendix A.

**Figure 2 genes-12-01815-f002:**
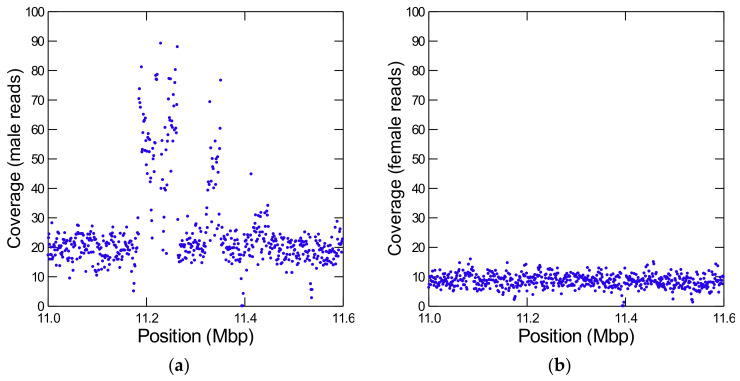
Detection of a Y-linked segmental duplication from an autosome. The figure shows the coverage of Illumina reads from males (**a**) and females (**b**) in a part of the autosomal scaffold CH964272. Note the huge increase in male coverage between positions ~11.2–11.4 Mbp, which is absent in the female reads. It corresponds to a segmental duplication of at least 265 kb from CH964272 to the Y chromosome. The coverage is very high and irregular, indicating many partial duplications and deletions inside the Y-chromosome. Each dot is the average depth of 1 kb interval (to reduce scattering). In the raw data, some regions have at least ten copies.

**Figure 3 genes-12-01815-f003:**
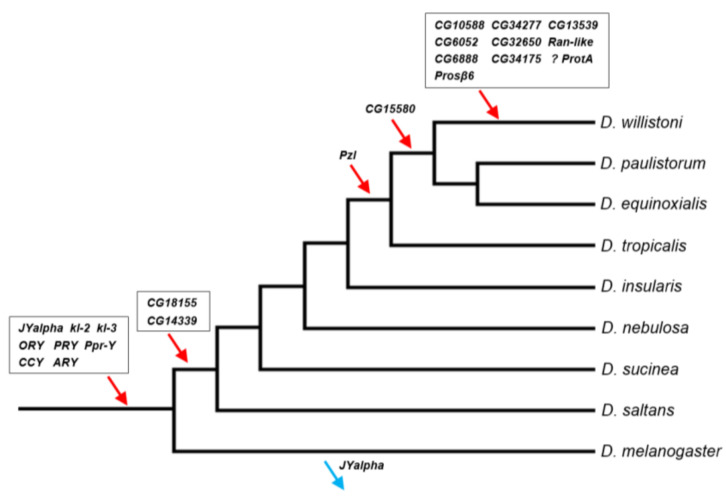
Timeline of gene acquisitions by the Y chromosome of the *D. willistoni* lineage. Gene location (Y-linkage vs. autosomal/X-linkage) was determined by PCR, except for the four genes with very similar autosomal and Y-linked copies (*CG6888*, *CG34175*, *ProtA* and *Prosβ6*; see Material and Methods). We inferred the direction of the movements (gene gains, red arrows; gene losses, blue arrows) by synteny and parsimony [14,39]. The eight genes on the basal branch were already known to be Y-linked in *D. willistoni. JYalpha* is not Y-linked in *D. melanogaster* due to a gene movement to an autosome within the melanogaster group [13]. There is some uncertainty regarding the *ProtA* gene, as it may have been acquired in the previous branch (at the same point of *CG15580*; Appendix A). Genes were labeled with the names of the *D. melanogaster* orthologs to facilitate cross-species comparisons. Phylogenetic relationships were taken from [51].

**Figure 4 genes-12-01815-f004:**
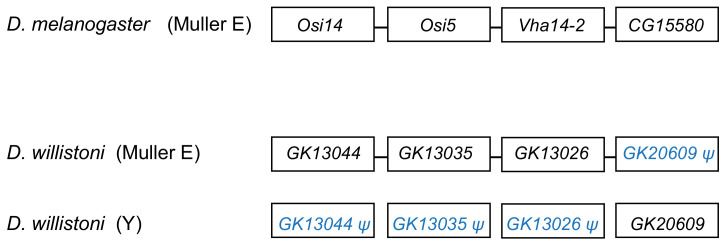
Fate of genes in the CH964272 segmental duplication. The segmental duplication copied 265 kb of autosomal sequence containing 25 genes to the Y chromosome of *D. willistoni* (for the sake of simplicity, the figure shows only four genes and omits several intervening ones). The Y-linked copy of *GK20609* (*D. melanogaster* ortholog: *CG15580*) remained functional, and the remaining 24 genes disappeared or became pseudogenes, whereas the opposite happened in the autosomal region. *CG15580*/*GK20609* and *Vha14-2*/*GK13026* are testis-specific genes in both species. The *D. willistoni* autosomal region is syntenic with *D. melanogaster* chromosome 3R (Muller E element). Gene order is unknown in the Y chromosome copy because it suffered duplications and deletions and the assembly is fragmented.

**Table 1 genes-12-01815-t001:** *D. willistoni* Y-linked genes found in this work.

D. willistoni Gene		D. melanogaster Ortholog			
Name	Copies ^1^	Name	Loc.	Expression ^2^	Predicted function/domains
*GK21041*	1 Y	*CG18155*	X	testis-specific	fatty acid biosynthetic process; mitochondrion location
*GK20609*	1 Y	*CG15580*	3R	testis-specific	leucine-rich repeat domain superfamily
*GK13929*	1 Y	*CG10588*	3L	testis + accessory gland	M16 metallo-endopeptidase protein present in spermatozoon
*GK28041*	1 Y	*CG32650*	X	testis-specific	protein of unknown function DUF4763
*GK27472*	1 Y	*CG13539*	2R	testis + accessory gland	protein of unknown function DUF1487
*YOgnWI030283*	1 Y	*CG34277*	3R	testis-specific	?
*GK21220*	1 Y	*CG6052*	3L	testis-specific	ATPase-coupled transmembrane transporter and lipid transporter. protein present in spermatozoon
*YOgnWI000172*	1 Y	*CG14339*	2L	testis-specific	cell division; regulation of mitotic sister chromatid separation.
*GK27406*	1 Y	*Piezo-like*	3Rhet	testis-specific	mechanosensitive ion channel
*GK28211*	1 Y	*Ran-like*	3L	testis-specific	GTP-binding protein involved in nucleocytoplasmic transport.
*GK18510*	10 Y, 1 A	*ProtA*	2L	testis-specific	protamine protein: DNA packing in sperm
*GK20591*	1 Y, 1 A	*CG6888*	3L	testis-specific	thioredoxin peroxidase; cell redox homeostasis; protein present in spermatozoon.
*YOgnWI018045*	1 Y, 1 A	*CG34175*	2L	testis-specific	?
*GK20618, GK20619*	4 Y, 1 A	*Prosβ6*	3L	widespread ^3^	component of the proteasome

^1^ Number of Y-linked and autosomal full-length copies (with 95% or higher nucleotide identity). ^2^ FlyBase data. Expression deemed as testis specific when expression in other organs is absent or very weak. ^3^ The source *D. willistoni* gene (*YOgnWI012342*) is testis-specific (Supplementary Material; Appendix A).

**Table 2 genes-12-01815-t002:** Y-linked segmental duplications in *D. willistoni*.

Autosomal Source					Y-Chromosome	
Chr.	Scaffold	Location (kb)	Size (kb)	Genes	Functional Genes	ψ Genes
E	CH964272	11,183–11,448	265	25	1 (*GK20609*)	12
C	CH963850	1340–1402	62	10	0	10
B	CH963913	3149–3452	303	14	2 (*GK18510*, *GK20618*)	10
B	CH963857	10,827–10,901	74	9	1 (*YOgnWI018045*)	4
total	-	-	704	58	4	36

## Data Availability

The data presented in this study are available in the Appendix A.

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
