# Peer review of "New Genes in the Drosophila Y Chromosome: Lessons from D. willistoni"

_genes, 2021, doi:10.3390/genes12111815_

Round 1

Reviewer 1 Report

General Comments

This is a nice paper, with some interesting findings on the unusual properties of the Y chromosome in Drosophila willistoni and its relatives. I am not an expert on genomic analysis, and therefore have to take the technical aspects of the paper on trust. The senior author has an excellent record in this regard, of course.

I do have some comments on the evolutionary aspects of the paper, and some minor matters of presentation (see Specific Comments below for these).

One major criticism is that they state several times (including in the Abstract) that the acquisition of new genes by the Drosophila Y is the major feature of its evolution, and they contrast this with the standard model of Y chromosome degeneration. While I understand what they mean, I think this is confusing for readers who are not familiar with this area. The point is that the ancestral Y of Drosophila has apparently lost all of its genes, in contrast to (say) the eutherian mammalian Y that has retained a number of them (mostly ones which were derived from the autosome that fused to the Y after the marsupial-eutherian split – there are only 3 genes remaining that were on the ancestral Y common to both lineages, I believe).

This loss happened a long time ago, so of course we can’t see evidence for it in modern Drosophila species. But the neo-Y of D. pseudoobscura has also completely degenerated over 13 million years, and that of D. miranda is well on its way after about 1.5 million years, so it is misleading to give the impression that degeneration is not going on in Drosophila, especially as their new data shows that many of the segmentally duplicated genes on the willistoni Y are now pseudogenes. Many would feel that the complete loss of several thousand genes from a chromosome is a more spectacular phenomenon than the transfer of a few genes onto another chromosome.

In addition, they downplay the evidence for transfers of genes from Y chromosomes to autosomes, and don’t mention the translocation of the ancestral Y to the dot chromosome in the D. pseudoobscura lineage and movements of Y genes to autosomes in other lineages (e.g. Bracewell & Bachtrog 2020 MBE 12: 494).

They need to give a more balanced presentation of these issues, I feel.

Their referencing style is inconsistent between different references (e.g. capitalization of first letter of journal title, full stops after abbreviated titles).

Some minor corrections to the writing are needed (see Specific Comments).

Specific Comments

l.42 ‘originating a..’ is not correct English- ‘leading to the evolution of a..’ is better. Also, they don’t explain why repetitive sequences should accumulate on a non-recombining chromosome- this is quite well understood in population genetic terms (e.g. Charlesworth et al. 1994 Nature 371:215).

l.60 ‘assemblies’ not ‘assembly’

l.66 Do they mean ‘the Y chromosome of D. willistoni’?

l.67 ‘divergence from’ not ‘divergence to’; ‘belongs’ not ‘belong’

l.71 ‘by’ not ‘in’

l.101 ‘onto’ not ‘to’

l.110 ‘lost recombination’ is a bit vague; they need to state explicitly that male Drosophila lack recombinational exchange between homologues, so that a Y or neo-Y chromosome automatically  fails to recombine.

l.131 and later ‘wingei’ not ‘winge’

l.208, 209 and later ‘left-hand’, ‘right-hand’ not ‘left’, ‘right’

l.266 ‘largest gene’?

l.293 It is unclear to me why genes involved in arms races should not survive long.

l.311 ‘it came to’ not ‘it was called to’

l.334 ‘created’ not ‘originated’

l.336 ‘what’ not ‘which’

l.342-344 This is unclear; I think what they are saying is that testis-specific expression seems to be necessary but not sufficient for the retention of genes on the Y.

l.348 ‘duplication’ not ‘duplications’

l.353 add ‘a’ before ‘segmental’; ‘It is known that’ not ‘It turns out that’

l.403 add ‘that’ before ‘we’

l.414-416 This is apparently based on one movement off the Y in D. melanogaster. The analysis is misleading because they did not look at other groups, like the pseudoobscura and affinis lineages, which give a rather different picture (see General Comments). Either they should use all available information or not bother with a quantitative analysis. It’s also not clear how they got the p value and confidence intervals.

l.421-423 They are setting up a straw man here. It’s been known for a long while that genes can move onto the Y and survive (it’s happened in mammals as well, of course). This doesn’t undermine the validity of the model of degeneration of genes on ancestral Y chromosomes. As modeling by people like Bachtrog (2008 Genetics 179: 1513) has shown, once most genes are gone from the Y, the selective interference forces that cause degeneration become weak, so it’s in fact quite hard to understand why Y’s have become completely degenerate, not why some genes can survive on them.

l.446 ‘more frequent’ not ‘increased’

l.458 ‘us’ not ‘we’

l.546 But this would apply to any kind of duplication, so it’s not clear why the Y should be favored. One possibility, which I don’t think they mention, is that the Y is largely gene free, so there is greater likelihood that material can be inserted without disrupting the structure or regulation of other genes.

l.552 This is not exactly what Fisher said; he showed that mutation with a sexually antagonistic effect is more likely to spread if it is linked to a male determining fact. He wasn’t talking about gene movements or origins.

l.553-557 I think this point could be made more concisely. If a gene is testis-specific in its expression, it cannot affect female fitness, so there is no selective benefit for its removal to the Y from sexual antagonism.

Reviewer 2 Report

Distinguishing between selective vs. neutral processes in shaping the current state of highly degenerated Y chromosome has been a major topic of research. While the sexual antagonism theory is the main hypothesis concerning the evolution of suppressed recombination, empirical evidence supporting this hypothesis in most system is lacking. In their manuscript, Ricchio et al, aim at identifying new Y-linked genes in D. willistoni , infer duplication event and its mechanism and discuss the role of natural selection vs. drift in shaping the current gene content of the Y chromosome in D. willistoni. I found the style of writing very engaging with clear and detailed description of methods and results and a thorough discussion. Below I have a few minor comments which I hope can help to improve the manuscript.

Lines 86 to 96: I think it is worthwhile to explain the neutral processes a bit more clearly. I checked the SI in reference 13. Here, despite what is currently stated on line 95, it is written that house-keeping or female-specific genes might be transferred to Y but there will be selection to keep the original autosomal copy since females need those genes. Due to the importance of random processes, it can be very useful to explain this process a bit more clearly.

Lines 101 to 117: I think it is clearer to end the introduction with the goals of the current manuscript. It might be more suitable to move this last paragraph to the section of the introduction where the mechanism of new gene gains is explained, that is lines 71 to 82. I’d therefore suggest to move line 65 “In this paper, we describe the identification of the protein-coding genes of D. willistoni” to the end of the introduction and end by mentioning that the role of selection vs. drift is discussed.

Line 259: It is stated that 11 out of 14 genes have been automatically annotated during the sequencing of the D. willistoni genome by Clark et al and the remaining three by Yang et al. It would be worthwhile to mention the exact contribution of the current manuscript. While these genes were previously annotated, was it their Y-linkage that was not detected? 

Line 272: It is stated that all 14 genes were acquired by gene duplications. Could the authors explain in more detail how this was inferred in the current study?
